Genome-wide identification of the SWEET gene family mediating the cold stress response in Prunus mume

Wen Zhenying
Li Mingyu
Meng Juan
http://orcid.org/0000-0002-8487-5896 Li Ping
Cheng Tangren
Zhang Qixiang
Sun Lidan sld656@126.com
Beijing Key Laboratory of Ornamental Plants Germplasm Innovation & Molecular Breeding, School of Landscape Architecture, National Engineering Research Center for Floriculture, Beijing Laboratory of Urban and Rural Ecological Environment, Beijing Forestry University , Beijing , China
Eamens Andrew
Electronic publication date: 2022 May 3
Publication date: 2022
Volume: 10
Electronic Location ID: e13273
Received 2021 Nov 8; Accepted 2022 Mar 23
Copyright: © 2022 Wen et al.
Copyright year: 2022
Copyright holder: Wen et al.
License: This is an open access article distributed under the terms of the Creative Commons Attribution License, which permits unrestricted use, distribution, reproduction and adaptation in any medium and for any purpose provided that it is properly attributed. For attribution, the original author(s), title, publication source (PeerJ) and either DOI or URL of the article must be cited.
License URL: https://creativecommons.org/licenses/by/4.0/

Keywords: Prunus mume, SWEET gene family, Gene expression, Cold response

Funding: Forestry and Grassland Science and Technology Innovation Youth Top Talent Project of China 2020132608 National Key Research and Development Program of China 2018YFD1000401 National Natural Science Foundation of China 31870689 This work was supported by Forestry and Grassland Science and Technology Innovation Youth Top Talent Project of China (No. 2020132608), the National Key Research and Development Program of China (2018YFD1000401), and the National Natural Science Foundation of China (No. 31870689). The funders had no role in study design, data collection and analysis, decision to publish, or preparation of the manuscript.

==============================
The Sugars Will Eventually be Exported Transporter (SWEET) gene family encodes a family of sugar transporters that play essential roles in plant growth, reproduction, and biotic and abiotic stresses. Prunus mume is a considerable ornamental wood plant with high edible and medicinal values; however, its lack of tolerance to low temperature has severely limited its geographical distribution. To investigate whether this gene family mediates the response of P. mume to cold stress, we identified that the P. mume gene family consists of 17 members and divided the family members into four groups. Sixteen of these genes were anchored on six chromosomes, and one gene was anchored on the scaffold with four pairs of segmental gene duplications and two pairs of tandem gene duplications. Cis-acting regulatory element analysis indicated that the PmSWEET genes are potentially involved in P. mume development, including potentially regulating roles in procedure, such as circadian control, abscisic acid-response and light-response, and responses to numerous stresses, such as low-temperature and drought. We performed low-temperature treatment in the cold-tolerant cultivar ‘Songchun’ and cold-sensitive cultivar ‘Zaolve’ and found that the expression of four of 17 PmSWEETs was either upregulated or downregulated with prolonged treatment times. This finding indicates that these family members may potentially play a role in cold stress responses in P. mume. Our study provides a basis for further investigation of the role of SWEET proteins in the development of P. mume and its responses to cold stress.

Introduction

Sucrose is the main carbohydrate in most plants; it is synthesized in the leaves during photosynthesis and then transported by phloem sap to storage organs, such as roots, stems, flowers, seeds and fruits (Rennie & Turgeon, 2009; Lemoine et al., 2013). Sucrose provides energy and carbon sources for plants and acts as an important signal and resistance molecule that participates in the normal growth of higher plants (Chen et al., 2015). However, sucrose must be assisted by appropriate sugar transporters as it cannot be transported independently to the storage organs (Ainsworth & Bush, 2011). At present, three transporter families have been identified as essential sugar transporters: monosaccharide transporters (MSTs), sucrose transporters (SUTs), and Sugar Will Eventually be Exported Transporters (SWEETs) (Chen et al., 2010, 2015; Eom et al., 2015). Of these three families, SWEETs were the final gene family to be uncovered and were first identified by Chen et al. (2010) in Arabidopsis. SWEET proteins act as sugar transporters that mediate the inflow or outflow of phloem parenchyma sugar into the phloem apoplast (Slewinski, 2011; Braun, 2012; Chen, 2014). Unlike the SUT and MST families, which require energy to transport sugar across the plasma membrane (Maynard & Lucas, 1982; Lemoine, 2000), SWEET proteins promote the diffusion of sugar across concentration gradients at the cellular membrane or vacuolar membrane, regardless of the proton gradient or pH of the cellular environment (Chen et al., 2012, 2015).

SWEET proteins are characterized by conserved MtN3_saliva (MtN3_slv) transmembrane (TM) domains (Chen et al., 2012), also known as PQ-loop repeats (Eom et al., 2015; Feng & Frommer, 2015). SWEETs in eukaryotes commonly consist of seven transmembrane helices (TMHs), which contain a pair of 3-TMH repeats detached by an added helix (Xuan et al., 2013), and this structure has been described as the “3-1-3” TM SWEET structure (Chen et al., 2010). In contrast to the structure of eukaryote SWEET proteins, prokaryote SWEET proteins, known as SemiSWEETs, are composed of only three TMHs (Xuan et al., 2013). In eukaryotes, proteins that contain 6 or 7 TMHs are prevalent, but SemiSWEETs with 3 or 4 TMHs have also been detected in plant genomes. In a study of SWEET genes from 25 plant genomes, 140 of the 411 SWEET sugar transporters identified were classed as being a semiSWEET; with all of the identified semiSWEETs either lacking the first or second 3-TM domain or which only existed in a partial form (Patil et al., 2015). This data therefore demonstrates that the presence of semiSWEETs in higher plant genomes is not unusual, and further, that SWEETs may in actual fact have formed by direct fusion from SemiSWEETs (Jia et al., 2017). In addition, a novel extraSWEET protein consisting of 14 and 15 TMHs has been reported from Vitis vinifera (Patil et al., 2015) and Oryza punctata (Jia et al., 2017); it is speculated that this extraSWEET may have formed from the duplication of a 7 TMH SWEET gene in these two species. Recent research on 3,249 SWEET proteins also identified a superSWEET with >18 TMHs in oomycetes, which carry 5–8 repeats of a semiSWEET (Jia et al., 2017). According to phylogenetic analysis, the SWEET genes in Arabidopsis can be divided into four clades: Clade I (SWEET1–3) and Clade II (SWEET4–8) mainly transport glucose, while Clade I SWEETs also have the ability to transport hexose (Chen et al., 2010; Lin et al., 2014). Clade III members (SWEET9–15) mainly transport sucrose (Chen et al., 2012; Eom et al., 2015), and Clade IV members (SWEET16–17), which are located on the tonoplast membrane, mainly transport fructose (Eom et al., 2015). The phylogenetic relationships of the SWEET genes described hereafter are all based on results from Arabidopsis.

Advances in whole-genome sequencing have enabled genome-wide identification of SWEET genes in numerous species. These include important crops, fruits and vegetables, such as rice (Oryza sativa) (Yuan & Wang, 2013), sorghum (Sorghum bicolor) (Mizuno, Kasuga & Kawahigashi, 2016), soybean (Glycine max) (Patil et al., 2015), apple (Malus domestica) (Wei et al., 2014), grape (Vitis vinifera) (Chong et al., 2014), banana (Musa acuminate) (Miao et al., 2017), tomato (Solanum lycopersicum) (Feng et al., 2015), rapeseed (Brassica napus) (Jian et al., 2016), potato (Solanum tuberosum) (Li et al., 2020) and valencia sweet orange (Citrus sinensis) (Yao et al., 2021). Additionally, many SWEET genes have been confirmed to play diverse and complex roles in physiological processes, such as nectar secretion (Ge et al., 2000; Lin et al., 2014), pollen development (Sun et al., 2013), senescence (Quirino, Normanly & Amasino, 1999), and seed filling (Sosso et al., 2015). Moreover, SWEET genes are also involved in biotic and abiotic stress responses (Yuan & Wang, 2013), including the reaction of plants to stress at low temperatures. For example, overexpression of AtSWEET16 and AtSWEET17 increases cold tolerance (Chardon et al., 2013; Klemens et al., 2013; Guo et al., 2014); overexpression of AtSWEET4 increases plant biomass and its resistance to frost (Chong et al., 2014; Liu et al., 2016); and AtSWEET11 and AtSWEET12 are involved in responses to cold or dehydration stress (Le Hir et al., 2015; Durand et al., 2016). AtSWEET15 is also known as SAG29 (where SAG stands for senescence-associated gene); however, its transcription level gradually increases at low temperature, high salinity, and drought during natural leaf senescence (Quirino, Normanly & Amasino, 1999). Cold stress significantly inhibits the expression of CsSWEET2, CsSWEET3, and CsSWEET16 in Camellia sinensis (tea plant), while the expression of CsSWEET1 and CsSWEET17 increases sharply (Yue et al., 2015). A functional study of CsSWEET16 in C. sinensis revealed that it is located in the vacuolar membrane, and furthermore, in transgenic Arabidopsis plants, CsSWEET16 expression regulates cold resistance (Wang et al., 2018). The transcriptional activity of many SlSWEET genes increases under low-temperature stress in tomato (Feng et al., 2015). Studies have shown that expression of the MaSWEET gene in banana is upregulated in response to low temperature, salt, and osmotic stress (Miao et al., 2017). Using genome-wide analysis of the BoSWEET gene in Brassica oleracea var. capitata (wild cabbage), five possible candidate genes were found to promote sugar transport and thereby enhance chilling tolerance of wild cabbage (Zhang et al., 2019).

Prunus mume is a traditional flower native to southwest China and the middle and lower reaches of the Yangtze River. In northern China, low temperatures severely limit the growth and distribution of this species. Although SWEET sugar transporters have been associated with responses to cold stress in other species, little is known about the role of PmSWEETs in cold responses in P. mume. This study aims to conduct a genome-wide analysis of the SWEET gene family in P. mume, with a specific focus on SWEET gene transcriptional responses to cold stress, providing a starting point to perform a detailed study of the potential functional roles of SWEET gene family members in P. mume.

Materials and Methods

Plant genomic resources

To explore the phylogeny of the SWEET genes in P. mume and other species, we downloaded SWEET proteins from two model plants (Arabidopsis thaliana and Oryza sativa, representing dicotyledons and monocotyledons, respectively) and eight other Rosaceae species. The protein sequences of 17 AtSWEETs and 21 OsSWEETs were downloaded from the TAIR 10 database (http://www.arabidopsis.org/) and TIGR (http://rice.plantbiology.msu.edu/), respectively. The P. mume genome sequence and annotation files were obtained from the P. mume genome project (https://github.com/lileiting/prunusmumegenome); the genomes of eight other Rosaceae species, including Malus domestica (Daccord et al., 2017), Prunus avium (Shirasawa et al., 2017), Prunus persica (Verde et al., 2013), Prunus yedoensis (Baek et al., 2018), Pyrus communis (Linsmith et al., 2019), Rosa chinensis (Raymond et al., 2018), Prunus salicina (Liu et al., 2020), and Prunus armeniaca (Jiang et al., 2019), were downloaded from the Genome Database for Rosaceae (https://www.rosaceae.org/).

Identification of SWEET genes in P. mume and other species

The hidden Markov model (HMM) profiles of the MtN3_slv domain for the SWEET gene family (PF03083) were downloaded from the Pfam database (http://pfam.xfam.org/) and used as queries to search for SWEET proteins in the proteomes of P. mume and other species with HMMER software (version 3.1b2, http://hmmer.org/) (Finn et al., 2015). To ensure confidence, the E-value cutoff was set at 10−5. Then, all putative SWEET proteins were screened to confirm the presence of the MtN3_slv domain by SMART (http://smart.embl-heidelberg.de/), the Pfam database (http://pfam.xfam.org/) and NCBI-CDD (https://www.ncbi.nlm.nih.gov/cdd), and sequences with MtN3_slv domain were retained.

The SWEET genes were named based on their location information in the P. mume genome. In addition, the number of amino acids, molecular weight (MW) and isoelectric point (pI) were calculated using the online ExPASy program (https://www.expasy.org/). The distributions of TM helices were predicted by use of the TMHMM Server v. 2.0 (https://services.healthtech.dtu.dk/service.php?TMHMM-2.0).

Phylogenetic and conserved domain analysis

To examine the phylogeny between SWEET genes in P. mume and other species, alignment of full-length SWEET protein sequences from three species (P. mume, A. thaliana, and O. sativa) and eight Rosaceae species was performed by using MAFFT software with the FFT-NS-1 strategy (Katoh & Standley, 2013). Subsequently, maximum likelihood (ML) phylogenetic trees were constructed using FastTree (version 2.1.11) (Price, Dehal & Arkin, 2010) with default parameters. Then, iTOL v4.0 (https://itol.embl.de/itol.cgi) (Letunic & Bork, 2019) and AI CS6 software were used to annotate and embellish the phylogenetic tree.

Conserved motif and gene structure analysis

The conserved motifs of each identified PmSWEET protein was predicted by MEME Suite Version 5.3.3 (https://meme-suite.org/meme/tools/meme) (Bailey et al., 2009), where the maximum number of motifs for the conserved domains was set to 10, motif width was set to 6–50 amino acids, and the residuals were designated as the default parameters. Gene structure data was extracted from the P. mume genome gff file, visualized using TBtools software (Chen et al., 2020), and then edited in AI CS6 software.

Chromosome location, duplication and synteny analysis

The location and chromosome length information of each PmSWEET gene was obtained from the gff file downloaded from the P. mume genome project (https://github.com/lileiting/prunusmumegenome). A chromosomal location figure was drawn using the online tool MG2C (http://mg2c.iask.in/mg2c_v2.0/). Gene tandem and segment replication events were analyzed using the Multiple Collinearity Scan Toolkit (MCScanX) and Circos in TBtools, respectively, with the default parameters. The synteny of the PmSWEETs across A. thaliana, P. armeniaca, and P. salicina was mapped using MCScanX in TBtools. The Ks and Ka values for duplicated gene pairs were calculated based on the coding sequence alignments using the Ka/Ks calculator in TBtools. According to two ordinary rates (λ) of 1.5 × 10–8 and 6.1 × 10–9 substitutions per site per year (Lynch & Conery, 2000; Blanc & Wolfe, 2004), the formula t = Ks/2λ × 10–6 Mya was used to calculate the divergence time.

Cis-acting element analysis of PmSWEET gene promoter regions

The upstream genomic sequence (2.0 kb) of each identified PmSWEET gene was retrieved from the genomic sequence data in TBtools and then submitted to the PlantCARE database (http://bioinformatics.psb.ugent.be/webtools/plantcare/html/) (Lescot et al., 2002) for cis-acting element analysis. We finally selected 12 elements, including those induced by hormones, such as methyl jasmonate (MeJA)-responsive, abscisic acid (ABA)-responsive, and stress-responsive elements; the stress-responsive factors included those involved in defense and stress, low temperature, and light. By combining these data with phylogenetic tree information (nwk file), the map was constructed by TBtools and edited by AI CS6 software.

PmSWEET genes expression analysis

To investigate the function of PmSWEETs involved in tissue development and cold tolerance, we used root, stem, leaf, flower bud and fruit data from RNA sequencing (Zhang et al., 2012) to analyze the PmSWEET expression patterns in different tissues and then used flower bud dormancy data from RNA sequencing of P. mume (‘Zaolve’) (Zhang et al., 2018) to analyze PmSWEET responses to low temperature from November to February. Furthermore, we explored the expression of SWEET gene family members in the stem of P. mume (‘Songchun’) in geographically distinct locations, including Beijing (BJ, N39°54′, E116°28′), Chifeng (CF, N42°17′, E118°58′) and Gongzhuling (GZL, N43°42′, E124°47′) and for three different periods of the year, including cold acclimation (October, autumn), the final period of endo-dormancy (January, winter), and deacclimation (March, spring) (Jiang, 2020). TBtools (Chen et al., 2020) was used to create the heatmap.

qRT–PCR analysis of PmSWEET genes

To examine the response of PmSWEET to low temperature, the annual branches of the cold-sensitive cultivar ‘Zaolve’ and the cold-tolerant cultivar ‘Songchun’ were collected. Before chilling treatment, the shoots were incubated overnight at 22 °C and then transferred to 4 °C for 0, 1, 4, 6, 12, 24, 48, and 72 h under long-day conditions (16-h light/8-h dark). The stems were collected immediately and transferred to liquid nitrogen until their longterm storage at –80 °C in readiness for RNA extraction. Each treatment had three biological replicates.

Total RNA of each sample was extracted using the RNAprep Pure Plant Plus Kit (Tiangen, Beijing, China). Complementary cDNA was synthesized using ReverTra Ace® qPCR RT Master Mix with gDNA Remover (Toyobo, Osaka, Japan). The specific primers were designed by Primer 3 (https://bioinfo.ut.ee/primer3-0.4.0/) based on the cDNA sequences (Table S1). The expression levels of PmSWEETs at low temperature were analyzed using quantitative real-time polymerase chain reaction (qRT–PCR) with a PikoReal real-time PCR system (Thermo Fisher Scientific, San Francisco, CA, USA) with SYBR® Green Premix Pro Taq HS qPCR kit (Accurate Biology, Changsha, China). The reactions were performed in a 10 μL volume, including 5.0 μL SYBR® Green Premix Pro Taq HS qPCR master mix, 0.5 μL each of forward and reverse primer, 1.0 μL of cDNA and 3.0 μL of ddH2O. The reactions were performed according to the following procedure: 95 °C for 30 s, followed by 40 cycles of 95 °C for 5 s and 60 °C for 30 s. Through the use of the PHOSPHATASE 2A (PP2A) gene of P. mume as the reference gene, the relative expression was calculated by using the delta-delta CT method (Livak & Schmittgen, 2001). Each qRT-PCR was conducted via the use of three biological replicates. The statistical analyses of ‘Zaolve’ and ‘Songchun’ were conducted independently using SPSS22.0, the one-way ANOVA analysis of variance was calculated by least significant difference (LSD) and Student–Newman–Keuls test with significant difference at level p = 0.05. GraphPad Prism6 software was used to draw the diagram.

Results

Identification of members of the Prunus mume SWEET gene family

A total of 17 nonredundant PmSWEETs were detected in the P. mume genome (sequence information is shown in File S1), and 175 SWEETs were detected in the eight other species of Rosaceae, including 16 SWEET genes in P. armeniaca, 19 in P. avium, 19 in P. persica, 19 in P. salicina, 16 in P. yedoensis, 21 in P. communis, 29 in M. domestica, and 36 in R. chinensis with rigorous filtering. All the newly identified SWEET genes were named according to their chromosome location (Table 1; Table S2). We determined that candidates with at least one MtN3_slv domain were “genuine” SWEET genes (domain architecture of PmSWEETs is shown in File S2). The number of amino acids, molecular weight (MW), and isoelectric point (pI) were calculated on the basis of the protein sequence of each identified SWEET. As exhibited in Table 1, the predicted PmSWEET proteins ranged from 105 (PmSWEET14) to 580 (PmSWEET8) amino acids in length, with relative molecular weights ranging from 15.96 kDa (PmSWEET11) to 63.43 kDa (PmSWEET8), and theoretical pIs ranging from 8.30 (PmSWEET4) to 9.76 (PmSWEET3). The MW and pI of family member PmSWEET14 could not be determined using this approach however due to the presence of four consecutive undefined amino acids (Table 1). Through prediction and analysis of TMHs of the 17 identified PmSWEETs, we found that these PmSWEET proteins were predicted to have 2–7 TMHs. Surprisingly, only seven members of the P. mume SWEET gene family were determined to possess standard 7 TMHs, most other SWEETs have fewer than 7 TMHs. Detailed location information of the TMHs is shown in Table S3 and Fig. S1.

Table 1 The PmSWEET gene family members in P. mume.

Name	Gene ID	Clade	CDS
(bp)	No. of amino acids	Molecular weight (kDa)	Theoretical pI	TMHs	No. of MtN3/saliva domain	Locus	
PmSWEET1	Pm007067	III	849	282	31.38	8.34	7	2	Pa2:21184396…21186332	
PmSWEET2	Pm008206	IV	759	252	27.74	8.50	7	2	Pa2:31718730…31721555	
PmSWEET3	Pm010330	I	1248	415	46.25	9.76	7	2	Pa3:3891190…3895205	
PmSWEET4	Pm011260	I	708	235	26.45	8.30	7	2	Pa3:9921623…9924001	
PmSWEET5	Pm013198	II	519	172	19.42	8.97	5	1	Pa4:2433448…2434735	
PmSWEET6	Pm015728	II	708	235	25.67	9.21	5	2	Pa4:21122646…21124537	
PmSWEET7	Pm017566	IV	735	244	26.99	9.14	7	2	Pa5:12327097…12328384	
PmSWEET8	Pm018875	III	1743	580	63.43	8.34	6	2	Pa5:20984940…20990591	
PmSWEET9	Pm019954	III	828	275	30.68	9.20	7	2	Pa6:436315…437664	
PmSWEET10	Pm021931	II	708	235	26.60	8.59	6	2	Pa6:12459796…12461199	
PmSWEET11	Pm022695	I	417	138	15.96	9.74	3	1	Pa6:19934418…19935334	
PmSWEET12	Pm022696	I	651	216	23.21	8.78	5	2	Pa6:19944525…19945680	
PmSWEET13	Pm024167	II	780	259	28.66	9.37	6	2	Pa7:10796671…10798904	
PmSWEET14	Pm024554	III	318	105	–	–	2	1	Pa7:13005181…13005663	
PmSWEET15	Pm024555	III	891	296	33.14	8.61	7	2	Pa7:13012731…13014646	
PmSWEET16	Pm024712	II	639	212	23.95	8.37	5	2	Pa7:13852243…13854234	
PmSWEET17	Pm030352	I	510	169	19.26	9.14	4	1	scaffold54:138478…139392	

Phylogenetic analysis and classification of SWEET genes

To better understand the evolution of homologous SWEET genes, we used the ML method to create a phylogenetic tree of all SWEET sequences from A. thaliana (model dicots), O. sativa (model monocots), and P. mume. According to previously reported AtSWEETs and OsSWEETs (Chen et al., 2010; Yuan & Wang, 2013), the 17 identified PmSWEETs were divided into four clades (i.e., Clade I, Clade II, Clade III, and Clade IV) (Fig. S2). To investigate the evolutionary relationships between PmSWEETs and the SWEETs of other species, a ML phylogenetic tree of SWEETs from 11 species, including 8 other Rosaceae species, was constructed. All members of the SWEET gene family in the 11 species were divided into four clades (Fig. 1). The largest clade was Clade III, which comprised five OsSWEET genes, seven AtSWEET genes, and 68 Rosaceae SWEET genes; the specific number of genes is shown in Table S4. The smallest clade was Clade IV, which consisted of only two A. thaliana SWEET genes, one O. sativa SWEET gene, and 18 Rosaceae SWEET genes (Table S4), a finding which shows that the SWEET genes are not evenly distributed across the four constructed clades. The numbers of genes in Clade I, II and III varied greatly, suggesting that the SWEET gene family expanded, especially in Clades I, II and III, during Rosaceae evolution. The SWEETs of Rosaceae were distributed uniformly across each small clade, whereas SWEETs from O. sativa tended to cluster together. The PmSWEETs, PpSWEETs, and PavSWEETs were clustered together and had similar distributions in the phylogenetic tree.

Figure 1 Phylogenetic tree of SWEET sequences from P. mume and other plant species.

Clades I, II, III, and IV are indicated by blue, indigo, orange and pale yellow branch lines, respectively. At, A. thaliana; Os, O. sativa; Pa, P. armeniaca; Pav, P. avium; Pc, P. communis; Pm, P. mume; Pp, P. persica; Ps, P. salicina; Py, P. yedoensis var. nudiflora; Md, M. domestica; Rc, R. chinensis.

Conserved motif and gene structure analysis

To explore the sequence features of PmSWEET proteins, MEME software and TBtools were used to predict and draw conserved domains. As a consequence, 10 distinct motifs were detected in SWEET proteins (Fig. 2B), and a schematic diagram of PmSWEET protein motifs is shown in Fig. S3. The number of PmSWEETs motifs was quite distinct, ranging from 1 to 7. Of them, 12 PmSWEETs contained more than four motifs, 4 PmSWEETs harbored four motifs, and PmSWEET14 contained only one motif. Motifs 1, 2, 3, 4 and 6 were highly conserved and present in 15 PmSWEET, 13 PmSWEET, 16 PmSWEET, 11 PmSWEET and 12 PmSWEET proteins, respectively; while motifs 7, 8 and 10 were relatively unique and existed in only 4 PmSWEET, 2 PmSWEET and 2 PmSWEET proteins, respectively. Intriguingly, aside from some unusual proteins, SWEET members of the same clade had similar conserved motifs, suggesting that they might have similar functions.

Figure 2 Phylogenetic relationship, conserved motif and gene structure analysis of PmSWEET genes.

(A) The ML phylogenetic tree of PmSWEET genes. The SWEET genes were grouped into four clades, and blue, purple, red, and green represents Clades I, II, III, and IV, respectively. (B) The motif composition of PmSWEET proteins. Ten motifs were displayed in different colored Rectangles. Motif1, GVVWFLYGLLKKDLFIAIPNGLGFJLGLVQLILYAIYR; Motif2, TKKRSLIVGIJCIVFNIIMYASPLTIMKLVIKTKSVEYMPFYLSLFLFLN; Motif3, LVITINGFGAVIELIYJAIFIIYAPKKKRKKI; Motif4, APVPTFYRIIKKKSTEEFQSVPYVAALLN; Motif5, WYGMPFVHPDN; Motif6, FGILGNIISFLLFL; Motif7, STNWDDDD; Motif8, PMTTLKRIMKKNEFTEQYLSGIPYLMT; Motif9, AMLWLYYGLLKPN; Motif10, NCZGCKDQYQHPQKCCKE. Detailed information is shown with logos obtained from the MEME Suite website in Fig. S3. (C) Exon-intron organization of PmSWEET genes. Green and black correspond to exons and introns, respectively.

To elucidate the structural characteristics of the PmSWEETs, the exon-intron structure was further analyzed. As shown in Fig. 2C, PmSWEETs in Clade II (except PmSWEET10) contained four introns. PmSWEET1, PmSWEET9, and PmSWEET15 in Clade III had five introns, PmSWEET8 contained the largest number of introns (12 introns), while PmSWEET14 contained only one intron. All PmSWEETs in Clade IV had five introns. The number of introns in Clade I varied from just 2 to 10, PmSWEET17 had two introns, PmSWEET4 contained five introns, PmSWEET11 and PmSWEET12 contained three introns, PmSWEET3 had 10 introns. These results indicated that aside from some unique gene family members, genes clustered together generally exhibited similar gene structures.

Chromosomal distribution and tandem duplication of PmSWEET gene family members

According to gene location information, all 17 PmSWEETs were mapped, showing that 16 PmSWEETs were located on chromosomes, and one PmSWEET gene was located on scaffold54 (Fig. 3). PmSWEET genes were mostly distributed on chromosomes 6 and 7, which both contained four PmSWEET genes. Two genes each were distributed on chromosomes 2, 3, 4 and 5. PmSWEET11 and PmSWEET12 as well as the PmSWEET14 and PmSWEET15 pair were clustered into two tandem duplication events on chromosomes 6 and 7, respectively. Based on the above results, some PmSWEETs gene family members were putatively generated by gene tandem duplication.

Figure 3 Schematic representations of the chromosomal location of the PmSWEET genes.

The chromosome number is indicated on the top of each chromosome and/or scaffold. Scf54 indicates scaffold54. Green and red gene names indicate the two identified tandem duplicated gene pairs.

Segmental duplication and synteny of the PmSWEET gene family

Synteny analysis of PmSWEETs was performed using the Circos program of TBtools, four segmental duplication events, including PmSWEET1/PmSWEET14, PmSWEET5/PmSWEET8, PmSWEET6/PmSWEET9 and PmSWEET6/PmSWEET16 were detected, and further, each gene pair was located on a different chromosome, as shown with red lines in Fig. 4. This finding strongly suggests that some PmSWEETs were likely generated by gene segmental duplication. In addition, the selection pressure and divergence time of the duplication events were estimated by the Ka (nonsynonymous) and Ks (synonymous) substitution ratio. In the evolutionary process, the Ka/Ks ratio >1 indicates positive selection (adaptive evolution), a ratio = 1 indicates neutral evolution (drift), and a ratio <1 indicates negative selection (conservation). Only one pair of segmentally duplicated PmSWEETs, namely PmSWEET6 and PmSWEET9, had a Ka/Ks ratio of 0.45, which was significant, and indicated a synonymous change that has been selected during plant genome evolution. The differentiation period of the PmSWEET6 and PmSWEET9 gene pair was approximately 55.34 to 136.07 Mya.

Figure 4 The Circos figure for PmSWEET segmental duplication links.

The red lines indicate segmented duplicated gene pairs.

To further examine the specific retention of PmSWEETs, their collinearity relationship with AtSWEETs, PaSWEETs, and PsSWEETs were detected using the MCScanX procedure of TBtools. A total of 16 homologous gene pairs were detected in P. mume and A. thaliana. Similarly, 16 pairs of homologous genes between P. mume and P. armeniaca and 20 between P. mume and P. salicina were detected (Fig. 5; Table S5). The collinear complexity of P. mume with P. salicina was much higher than that with P. armeniaca and A. thaliana. These results suggested that P. mume was relatively distantly related to A. thaliana and P. armeniaca, but is more closely related to P. salicina.

Figure 5 Synteny of SWEET genes in different genomes of P. mume, A. thaliana, P. armeniaca and P. salicina.

(A) Synteny of PmSWEET and AtSWEET gene pairs. (B) Synteny of PmSWEET and PaSWEET gene pairs. (C) Synteny of PmSWEET and PsSWEET gene pairs.

Prediction analysis of Cis-acting elements within PmSWEETs gene promoters

To further investigate the possible regulatory mechanism of PmSWEETs in the process of growth or in plant defence mechanisms, in particular the response of a plant to abiotic stresses such as low temperature, we submitted the 2.0 kb upstream sequence from the translation start site of each PmSWEET gene to the PlantCARE database to search for the presence of specific cis-elements. The PmSWEET promoters comprised several conserved regulatory elements that respond to plant hormones and environmental stress, and 12 of these were analyzed further (Fig. 6; Table S6). Elements related to light response, anaerobic induction, and ABA response were widespread in the promoter areas of 17, 17 and 16 members of the P. mume SWEET gene family, respectively. According to the regulatory elements in their promoters, 14, 12, 11, 10, and 9 P. mume SWEET gene family members were sensitive to drought inducibility, MeJA, gibberellin, low temperatures and auxin, respectively. By combining these findings with the results of phylogenetic analysis, it was found that gene members of the same clade had similar cis-elements. These results indicated that PmSWEET genes were involved in the regulatory mechanisms of various stress responses.

Figure 6 Predicted cis-elements responding to plant growth regulation, hormone response, and stress response present in PmSWEET gene promoters.

Different colored boxes represent different elements and their positions in each PmSWEET promoter. The SWEET genes are classified into four clades, and blue, indigo, purple red, and green represent Clades I, II, III, and IV, respectively.

Expression pattern analysis of PmSWEETs

To investigate the role of PmSWEETs in development and response to low temperature, the expression patterns of family members in the roots, stems, leaves, flower buds, fruits (Zhang et al., 2012) and flower buds of different stages of dormancy (Zhang et al., 2018), were examined based on our RNA-seq dataset, and their RPKM values are shown in Tables S7 and S8. As illustrated in Fig. 7A, 14 of the PmSWEET genes were expressed in at least one tissue, whereas RNA-seq failed to detect the expression of three family members, namely PmSWEET5, PmSWEET10 and PmSWEET11. Among them, five PmSWEETs presented relatively higher expression levels in fruits (PmSWEET1, PmSWEET6, PmSWEET9, PmSWEET12 and PmSWEET17) and flower buds (PmSWEET3, PmSWEET13, PmSWEET14, PmSWEET15 and PmSWEET16). Two PmSWEETs showed higher expression levels in roots (PmSWEET4 and PmSWEET7) and stems (PmSWEET2 and PmSWEET8). Additionally, genes PmSWEET2, PmSWEET3, PmSWEET4, PmSWEET7, PmSWEET8, PmSWEET12 and PmSWEET13 were expressed in leaves, but their expression levels were low.

Figure 7 Expression profiles of PmSWEET genes in different tissues and different flower buds stage.

(A) Expression profiles of PmSWEETs in different tissues. (B) Expression profiles of PmSWEETs in the flower bud during dormancy. EDI, endo-dormancy I, November; EDII, endo-dormancy II, December; EDIII, endo-dormancy III, January; NF, natural flush, February. A 2-based log function conversion is performed on the expression amount, and then normalized by row using min-max method. The color scale on the right of the heat map refers to relative expression level, and the color gradient from blue to red shows an increasing expression level.

Most PmSWEETs were expressed during the flower bud dormancy period (except PmSWEET5 and PmSWEET16) as well as being expressed at specific stages of development (Fig. 7B). Ten PmSWEET genes exhibited specifically higher expressions in the natural flush (NF) stage (February), PmSWEET9 was preferentially expressed in the endo-dormancy I (EDI) stage (November), PmSWEET10 and PmSWEET12 showed the highest level of expression in the endo-dormancy II (EDII) stage (December); and PmSWEET1, PmSWEET3, PmSWEET6, PmSWEET12 and PmSWEET13 showed upregulated expression in the endo-dormancy III (EDIII) stage (January). Among these upregulated genes, eight PmSWEETs (PmSWEET6, PmSWEET7, PmSWEET10, PmSWEET11, PmSWEET13, PmSWEET14, PmSWEET15 and PmSWEET17) (Table S6) contained low temperature response elements within their putative promoter regions.

To further investigate the expression patterns of PmSWEETs under cold exposure, we analyzed the stems of the cold-tolerant cultivar P. mume ‘Songchun’ at three geographically distinct locations (Jiang, 2020), and their FPKM values are displayed in Table S9. The expression of six PmSWEET genes (PmSWEET5, PmSWEET6, PmSWEET11, PmSWEET14, PmSWEET16 and PmSWEET17) was not detected. Among the other 11 PmSWEET genes, seven PmSWEETs (PmSWEET1, PmSWEET2, PmSWEET3, PmSWEET4, PmSWEET7, PmSWEET8 and PmSWEET9) showed higher expression in spring (3.2~5.3 °C). PmSWEET13 expression was upregulated in autumn (6.1~7.9 °C) and winter in Beijing (–5.4 °C) and Chifeng (–11.4 °C), but downregulated in spring; the expression levels of PmSWEET10, PmSWEET12 and PmSWEET15 increased significantly in winter in Beijing (–5.4 °C) (Fig. 8A). Among these genes with upregulated expression, four PmSWEETs (PmSWEET7, PmSWEET10, PmSWEET13 and PmSWEET15) (Table S6) contained low-temperature response elements within their putative promoter regions. To compare the expression patterns of PmSWEETs during different times of the year, another heatmap was generated (Fig. 8B). As shown in Fig. 8B, PmSWEETs expression in the material sourced from the locations, Chifeng and Gongzhuling showed similar expression patterns at the same time of the year, while PmSWEETs expressed for the material sourced from the Beijing location showed higher expression in winter (Fig. 8B). This may be related to the latitude of the three geographical sampling locations, Gongzhuling has the highest latitude, followed by Chifeng and Beijing. There is little difference between the temperature in autumn and spring in the three sampling locations, however there is considerable difference in the winter temperature. In winter, the temperature in Beijing (–5.4 °C) is higher than that in the other two sampling locations (Gongzhuling is –22.8 °C, Chifeng is –11.4 °C), which may be the temperature that is required to induce the expression of some P. mume SWEET gene family members.

Figure 8 Expression profiles of PmSWEETs in stems in different seasons and regions.

(A) Expression profiles of PmSWEETs in stems of ‘Songchun’ in different regions (Beijing, Chifeng and Gongzhuling) and seasons (autumn, winter and spring). (B) Comparison of differential expression profiles of stems in Beijing, Chifeng and Gongzhuling during different seasons. A 2-based log function conversion is performed on the expression amount, and then normalized by row using min-max method. The color scale on the right of the heat map refers to relative expression level, and the color gradient from blue to red shows an increasing expression level. Aut, Autumn; Win, Winter; Spr, Spring; BJ, Beijing; CF, Chifeng; GZL, Gongzhuling.

Expression patterns of P. mume SWEETs under cold treatment

To investigate the role of PmSWEETs in response to cold stress, the expression patterns under imposed stress treatment temperature of 4 °C for 0, 1, 4, 6, 12, 24, 48 and 72 h were examined by qRT–PCR using the cold-sensitive cultivar ‘Zaolve’ and the cold-tolerant cultivar ‘Songchun’. We performed a qRT–PCR assay on the 17 identified P. mume SWEETs, but the expression of only 11 PmSWEETs was detectable by this approach, while the remaining 6 PmSWEETs (PmSWEET5, PmSWEET6, PmSWEET9, PmSWEET11, PmSWEET15 and PmSWEET16) were not detected, a finding that is consistent with the transcriptome data (Figs. 7 and 8). As displayed in Fig. 9, the changes in expression levels of the 11 SWEET genes in the two cultivars differed during the imposed cold stress treatment period. In the two assessed cultivars, the expression of three genes, PmSWEET2, PmSWEET7 and PmSWEET8, was reduced. In addition, the expression of PmSWEET13 was upregulated in both ‘Songchun’ and ‘Zaolve’, which rose approximately 11-fold after 6 h of cold treatment in ‘Songchun’, while rising approximately 9-fold after 1 h, and then increased nearly 80-fold after 72 h of cold treatment in ‘Zaolve’. One gene (PmSWEET3) changed only slightly in both ‘Songchun’ and ‘Zaolve’. Six genes (PmSWEET1, PmSWEET4, PmSWEET10, PmSWEET12, PmSWEET14, and PmSWEET17) exhibited different expression patterns in the two cultivars. Among these six genes, PmSWEET1 and PmSWEET12 were upregulated initially, then downregulated with increasing treatment duration in ‘Songchun’, while in ‘Zaolve’, there was no obvious change in the early treatment stages, but the expression of these two PmSWEETs increased considerably at 48 h and 72 h, respectively. PmSWEET4 and PmSWEET10 were dramatically downregulated in their level of expression with increased cold stress duration in ‘Songchun’, while the expression of these two PmSWEETs was upregulated within 6 h and then decreased with extended treatment in ‘Zaolve’. PmSWEET14 expression did not show an obvious change across the early stages of treatment, but was rapidly upregulated at 72 h in ‘Songchun’, and at 24 h in ‘Zaolve’, and then the expression level of PmSWEET14 decreased in ‘Zaolve’ with increasing treatment duration. The expression of PmSWEET17 was increased during the early stages of treatment, but then decreased with increased treatment duration in ‘Songchun’, while it was highly expressed only at 4 h in ‘Zaolve’.

Figure 9 Expression patterns of 11 PmSWEET genes under low temperature treatments.

The relative quantification method (2–ΔΔCt) was used to evaluate the transcript levels of 11 PmSWEET genes. Error bars are standard deviation of three biological replicates. The statistical analyses of ‘Zaolve’ and ‘Songchun’ were conducted independently using SPSS22.0, the one-way ANOVA analysis of variance was calculated by least significant difference (LSD) and Student–Newman–Keuls test, diﬀerent letters above the bars indicate significant diﬀerences (p = 0.05). Black letters indicate ‘Zaolve’, red letters indicate ‘Songchun’. GraphPad Prism6 software was used to draw the diagram.

Discussion

SWEET genes form a family of sugar transporters that play a role in the transportation of sugars, mainly sucrose, glucose and fructose (Chen et al., 2010, 2012; Feng & Frommer, 2015; Guo et al., 2014; Klemens et al., 2013; Le Hir et al., 2015), and due to this important role, SWEETs have been demonstrated to function in diverse physiological and biological processes in the growth and development of many plants as well as in the response of these plant species to biotic and abiotic factors (Lemoine et al., 2013; Li et al., 2017, 2018; Zhao et al., 2018). Previous studies have shown that SWEETs participate in cold stress responses in several plant species (Chardon et al., 2013; Klemens et al., 2013; Guo et al., 2014; Chong et al., 2014; Liu et al., 2016; Le Hir et al., 2015; Yue et al., 2015; Wang et al., 2018; Feng et al., 2015; Miao et al., 2017; Zhang et al., 2019). However, little is known about the potential roles of SWEET genes in the response of P. mume to cold stress. P. mume has a high ornamental value, and can blossom at lower temperatures; but different cultivars have different cold resistance, making it an ideal plant species for studying the mechanisms of how SWEET genes function in cold responses. Understanding the link between SWEET genes of P. mume and cold-resistance could provide insights into cold-resistance molecular breeding in the future. In this research, we identified a total of 17 SWEET genes in P. mume, the same number that is that present in Arabidopsis, and similar to the numbers in other species of Prunus, showing that SWEET genes are still relatively conserved in Prunus. The length of PmSWEET proteins ranged from 105 aa to 580 aa, and this range provides diversity in the number of TMHs (2–7). PmSWEETs, except for PmSWEET14, have a theoretical pI larger than 8.0. As an important parameter of proteins, pI is determined by the relative contents of amino acid residues at different pH values, which affects the stability, activity and function of a protein (Gasteiger, 2005). The pI of PmSWEET14 was not detected, which may be due to its short amino acid sequence.

By predicting TMH domains, we found that the number of TMHs encoded by PmSWEET genes ranged from 2 to 7 (Table 1). Fewer than seven TMHs in members of the SWEET gene family has also been reported previously in other plant species, including wheat (Gao et al., 2018; Gautam et al., 2019), walnut (Jiang et al., 2020), Kentucky bluegrass (Zhang, Niu & Ma, 2020) and soybean (Patil et al., 2015). To further validate the accuracy of our SWEET protein predictions, we submitted the protein sequence of each PmSWEET to the NCBI-CDD and SMART online tools to predict their conserved domains, and it was found that each assessed family member contained the MtN3_slv domain, and therefore, belonged to the SWEET gene family. This result also indicated that duplication and fusion, or genetic loss may have occurred to individual SWEET gene loci as part of the evolution of the P. mume genome. Similar to the case in other plant species (Chen et al., 2010; Yuan & Wang, 2013; Patil et al., 2015), PmSWEETs can be classified into four clades, and the number of SWEET genes members from 11 plant species ordered into Clade III was larger than that in the other three clades (Fig. 1), suggesting that Clade III may have expanded during genome evolution. Conserved motif analysis indicated that some special motifs only reside in some PmSWEET gene family members. For instance, motif 8 was only present in PmSWEET11 and PmSWEET17; and motif 10 was only present in PmSWEET3 and PmSWEET15. These results are consistent with those of other plant species, such as Arabidopsis (Chen et al., 2010), rice (Yuan & Wang, 2013), banana (Miao et al., 2017) and wheat (Gautam et al., 2019). Together, these studies have demonstrated that gene structural diversity and conserved protein motif divergence has performed a key role in the evolution of the SWEET gene family (Xu et al., 2012). More specifically, specific PmSWEETs harbored unique conserved motifs, implying that such family members may be responsible for the functional diversity of the P. mume SWEET gene family.

Gene duplication, including tandem and segmental duplication events, is the origin of gene family expansion as part of genome evolution in plants (Cannon et al., 2004; Ganko, Meyers & Vision, 2007). In this study, two pairs of PmSWEETs were detected as tandem duplications, and four pairs of PmSWEETs were identified to be the result of segmental duplications. This finding is consistent with those of other studies on SWEET duplication, including segmental and tandem duplications (Feng et al., 2015; Miao et al., 2017; Gao et al., 2018; Jiang et al., 2020).

The cis-elements in the promoter of a gene play an essential role in the regulation of gene expression, and therefore, gene function. All PmSWEETs contain at least one light-responsive and anaerobically induced cis-element, suggesting that these two elements have an essential role in regulating PmSWEET gene expression. Moreover, 10 PmSWEETs contained one or more low-temperature responsive cis-elements (Table S6), indicating that these PmSWEETs may play important roles in the response of a PmSWEET gene to cold stress. However, the exact regulatory role directed by these cis-elements in P. mume requires further research.

Studies have shown that under low-temperature stress, the soluble sugar content in plants increases, and sugar transporters maintain the balance of osmotic potential through the balance and distribution of sugar, thus improving the cold tolerance of plants (Yamada et al., 2010). Numerous studies have also verified that SWEETs are involved in maintaining sugar homeostasis in plant organs and promoting plant adaptation to low temperatures (Seo et al., 2011; Chardon et al., 2013; Klemens et al., 2013; Chandran, 2015; Le Hir et al., 2015; Miao et al., 2017; Wang et al., 2018; Zhang et al., 2019; Zhang, Niu & Ma, 2020). Transcriptome analysis showed that PmSWEETs were differentially expressed in different tissues and during dormancy release and cold acclimation. PmSWEET5 expression was not detected in any tissue/organ that was assessed in this study, indicating that its expression may be highly varietal, spatially and temporally specific. Some PmSWEETs had specific expression patterns in different organs (Fig. 7A). For example, PmSWEET10 expression was detected only in ‘Zaolve’ flower buds at dormancy (stage EDII) and ‘Songchun’ stems of those P. mume plants taken from the Beijing winter sampling site. Furthermore, PmSWEET16 expression was detected only in P. mume flower buds, which indicates that these two PmSWEETs are expressed only in specific tissues, cultivars, or environmental conditions, with such organ-specific expression previously observed in wheat (Gao et al., 2018; Gautam et al., 2019), walnut (Jiang et al., 2020), tea (Wang et al., 2018) and cabbage (Zhang et al., 2019). AtSWEET5, the homologue of PmSWEET10 and PmSWEET16, plays a key role in seed germination, and is expressed at different stages of pollen development (Engel, Holmes-Davis & McCormick, 2005). The results from expression studies of different organs indicate a role for PmSWEET10 and PmSWEET16 in pollen development, suggesting they might play a similar role to AtSWEET5. PmSWEET1, PmSWEET6, PmSWEET9, PmSWEET12 and PmSWEET17 were strongly expressed in fruit, to indicate that the proteins encoded for by these PmSWEET genes may regulate sugar allocation during fruit ripening. Such specific, and high levels of expression of SWEETs in fruits has also been found in pineapple (Guo et al., 2018), sweet orange (Zheng et al., 2014) and apple (Zhen et al., 2018), findings which collectively infer that SWEET proteins likely mediate an important role in fruit development and ripening. PmSWEET4 (Clade I) and PmSWEET7 (Clade IV) were strongly expressed in roots, this results had similar expression patterns to previous studies, those being that Clade IV SWEET genes are highly expressed in the root cortex and encode for proteins that function as fructose-specific uniporters in the root vacuole membrane (Guo et al., 2014).

The present results also show that most of the PmSWEET genes are expressed more strongly at different endo-dormancy stages in flower bud and fruit tissues than in other tissues and that these genes are differentially expressed during flower development (Figs. 7A and 7B). Together, these results suggest that the P. mume SWEET gene family is closely associated with reproductive development and that different genes are specifically involved during different developmental stages. In rice, Arabidopsis and soybean, the expression of SWEET genes is also higher in reproductive tissues than in other tissues (Yuan et al., 2014; Patil et al., 2015). PmSWEETs also have different expression levels during dormancy release in flower buds (from November to February). Thus, we speculate that these PmSWEETs may participate in the cold reaction at low temperatures to protect the flower bud. In addition, some PmSWEETs were expressed more highly at colder temperatures in the spring (3.2~5.3 °C) and at approximately –5 °C in the winter (Fig. 8A). Together, this finding putatively suggests that these two temperatures may trigger their cold stress response and increase PmSWEET expression to reduce stress injury.

The qRT–PCR analysis suggested that six of 17 PmSWEET genes (PmSWEET5, PmSWEET6, PmSWEET9, PmSWEET11, PmSWEET15, and PmSWEET16) were not expressed in the stem, which was consistent with the transcriptome data. PmSWEETs were activated by low temperature (4 °C) and increased or decreased in expression with the extension of treatment time (Fig. 9). The expression levels of five PmSWEETs (PmSWEET2, PmSWEET4, PmSWEET7, PmSWEET8, and PmSWEET10) in ‘Songchun’ and three PmSWEETs (PmSWEET2, PmSWEET7, and PmSWEET8) in ‘Zaolve’ decreased with increasing treatment duration (Fig. 9), which suggested that these genes might be negatively regulated by low temperatures and result in increased cold sensitivity. The expression levels of two PmSWEETs (PmSWEET13 and PmSWEET14) in ‘Songchun’ and three PmSWEETs (PmSWEET1, PmSWEET12, and PmSWEET13) in ‘Zaolve’ increased with prolonged treatment (Fig. 9), which suggested that these genes might be positively regulated by cold stress responses and increase the cold sensitivity of P. mume. The discrepancy in expression patterns between PmSWEET1, PmSWEET4, PmSWEET10, PmSWEET12, PmSWEET14 and PmSWEET17 is potentially due to genetic differences between ‘Songchun’ and ‘Zaolve’.

Conclusions

In summary, our study is the first to perform genome-wide identification and characterization of SWEETs in P. mume, including chromosomal location, duplicated gene identification, gene structure analysis, phylogenetic relationships and conserved motifs. In addition, the expression profiles of the PmSWEET genes in different tissues and geographic locations were also examined based on the RNA-seq data. Furthermore, the expression profiles of these PmSWEET genes under cold stress conditions were analyzed by qRT–PCR assay. Our results could provide important information for further research on the biological functions of PmSWEETs.

Supplemental Information

Supplemental Information 1 Supplemental tables.

Click here for additional data file.

Supplemental Information 2 Schematic representation of PmSWEET proteins.

Colored boxes indicate TMs. The IBS1.0 software was used to drawing diagram (Liu et al., 2015).

Click here for additional data file.

Supplemental Information 3 Phylogenetic trees of Arabidopsis thaliana, Prunus mume and Rice.

Click here for additional data file.

Supplemental Information 4 Schematic diagram of PmSWEET protein motifs.

Click here for additional data file.

Supplemental Information 5 Protein sequences of P. mume.

Click here for additional data file.

Supplemental Information 6 Domain architecture of PmSWEETs.

Click here for additional data file.

Supplemental Information 7 ’Zaolve’ qRTPCR raw data.

Expression analyses of 11 PmSWEETs in P. mume ’Zaolve’ exposed to 4 °C for different times (0/1/4/6/12/24/48/72 h), where 0 h indicates control.

Click here for additional data file.

Supplemental Information 8 ’Songchun’ qRTPCR raw data.

Expression analyses of 11 PmSWEETs in P. mume ’Songchun’ exposed to 4 °C for different times (0/1/4/6/12/24/48/72 h), where 0 h indicates control.

Click here for additional data file.

Additional Information and Declarations

Competing Interests

Author Contributions

Data Availability

The authors declare that they have no competing interests.

Zhenying Wen conceived and designed the experiments, performed the experiments, prepared figures and/or tables, and approved the final draft.

Mingyu Li analyzed the data, prepared figures and/or tables, and approved the final draft.

Juan Meng analyzed the data, prepared figures and/or tables, and approved the final draft.

Ping Li analyzed the data, authored or reviewed drafts of the paper, and approved the final draft.

Tangren Cheng conceived and designed the experiments, authored or reviewed drafts of the paper, and approved the final draft.

Qixiang Zhang conceived and designed the experiments, authored or reviewed drafts of the paper, and approved the final draft.

Lidan Sun conceived and designed the experiments, authored or reviewed drafts of the paper, and approved the final draft.

The following information was supplied regarding data availability:

The raw dates are available in the Supplemental Files.

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
