# Peer review of "Genome-wide identification of the SWEET gene family mediating the cold stress response in Prunus mume"

_PeerJ, doi:10.7717/peerj.13273_

## Round 0.1 · original submission · Major Revisions

Please very carefully read through the comments provided by the two peer reviewers of your manuscript. Each comment and/or concern raised by the two reviewers, especially those of Reviewer #1, must be specifically addressed in a very 'Major Revision' of your original submission.

More specifically, Reviewer #1 has raised multiple concerns about the quality of the science being reported on in your manuscript, and following my own review of your manuscript, I must admit that I too share the concerns of Reviewer #1.

Furthermore, the quality of the English language used through all sections of your manuscript is of a very low standard and must be improved considerably before your manuscript could be considered of a standard acceptable for publication consideration.

My major concerns are:

Materials and Methods - this entire section must be redone in order for the reader to properly comprehend the quality of the science being reported on. Currently, not enough experimental detail is supplied for the reader to accurately judge what quality of data was used for your bioinformatic analyses.

Furthermore, almost all sentences in the methods section are written in lab recipe / protocol format, and this format of writing is unacceptable. All Methods must be written in standard conversational text.

Results - the standard of the writing in this section too requires considerable improvement in order for the reader to be able to appreciate the impact of each set of results being reported on. Please very carefully rewrite this section of the manuscript.

Discussion - far too many statements are made on the small amount of wet lab experimentation reported on in the manuscript, largely the RT-qPCR data presented in Figure 8. A change in gene expression under an environmental change does not always equate to a functional role of that gene - especially when this is purely based on the functional role of an orthologous gene from a different species. Please therefore rewrite the Discussion to remove all instances of overstatements.

Please also more clearly state, in the Introduction and Discussion sections, why the SWEET gene family and cold stress were selected as the gene / environmental condition combination under analysis. You have failed to properly establish this relationship in your manuscript, so you must do this in the revised manuscript version.

There is considerable work which must be undertaken in this revision. Therefore, I do not expect to see a revised manuscript version submitted for sometime - many, many weeks to be specific. This is a VERY 'Major Revision' which I am requesting, so please do take great care and significant time preparing a vastly improved version of your study.

Please find my annotated PDF attached and use this as a reference when preparing your revised manuscript.

Kind regards,
Andrew

Reviewer 1 ·

Basic reporting

Zhenying Wen and coauthors present data on sequences with homology to SWEET proteins from the tree species Prunus mume and other Rosaceae species. The authors conduct phylogenetic analysis and expression analysis based on RNAseq and RT-qPCR analysis. In general, the comparison between a cold-sensitive and cold-tolerant accession of this interesting species opens the possibility for identifying factors responsible for cold tolerance. However, the presented study fails in the aspect to connect cold stress response and SWEET proteins and displays technical errors probably due to insufficient curation and presentation of used sequence material.

Experimental design

I think the major issue of the manuscript is that it is unclear if phylogenetic analysis of PmSWEETs have been done with complete and accurately curated sequences. Inspection of table 1 leaves the reader helpless to judge the quality of the used sequence material. After all, only six SWEETs possess seven transmembrane helices (TMHs), all other (except SWEET3) possess less. For all these SWEET types, seven TMHs have been reported in other species. Does this mean they are pseudogenes, unfunctional proteins or just incomplete sequences in Prunus mume? If so, any conclusion drawn from differential gene expression for a role of the respective transporter for cold tolerance would be not supported by their function.

Validity of the findings

I really think authors should manually inspect the extracted sequences and check for completeness and accuracy of the used sequence material. In this respect, I find it of paramount importance that the sequences of all analysed PmSWEETs been made available to the reader in a supplementary sheet and that both, coding sequences and TMHs be highlighted in these sequences. Without provision of the sequences, it impossible for the reader to comprehend the conducted analyses.

Some more comments:
The overall choice of SWEET protein family seems a bit arbitrary and needs better explanation and introduction already in the abstract since there is no real connection between SWEET activity and cold tolerance given. For example, other transporters mainly from the MST (monosaccharide transporter family) are much heavier involved in cold stress response and cold tolerance than SWEETs and would have probably been an equally important choice of in-depth analysis.
The statement in the abstract that SWEETs from different plant species divide into four group is not new (see Eom et al., 2015 Curr Op Plant Biology for Review) and does not explain the situation in P. mume.
Why do the authors classify “circadian control” as stress?
From the abstract, it is unclear why the mentioned accessions “Songchun” and “Zaolve” were chosen for comparative transcription analysis.

Reviewer 2 ·

Basic reporting

Wen et al. investigated the SWEET family in the Prunus m. and studied the transcript expression in different conditions. This work was well designed and conducted while I have a few comments, hopefully, to improve this manuscript.
Although the writing is smooth, I would recommend the authors check the grammar with Naïve speakers. In addition, probably change the title to “Genome-wide identification of the SWEET gene family mediating the cold stress response in Prunus mume”
1. Line 18-20: the sentence may be revised as “and medicinal values; however, the distribution…..by low temperature.” And delete the “there were no…..about SWEET gene family in P. mune”. BTW, please make sure to use Italic when referred the gene name.
2. Line 34-38/Line 337-343: is “composited” in … In addition, the product of photosynthesis is sucrose (part of sugars) as well as involved in the phenotypical-metabolic changes and interaction with plant hormones, so I would recommend to authors to include few references and revise the sentence.
Lastdrager, J., Hanson, J. & Smeekens, S. Sugar signals and the control of plant growth and development. J. Exp. Bot. 65, 799–807 (2014); Ruan, Y. L. Sucrose metabolism: gateway to diverse carbon use and sugar signaling. Annu. Rev. Plant Biol. 65, 33–67 (2014); Wang, J.Y., Alseekh, S., Xiao, T. et al. Multi-omics approaches explain the growth-promoting effect of the apocarotenoid growth regulator zaxinone in rice. Commun Biol 4, 1222 (2021). https://doi.org/10.1038/s42003-021-02740-8; Sulpice, R. & McKeown, P. C. Moving toward a comprehensive map of central plant metabolism. Annu. Rev. Plant Biol. 66, 187–210 (2015). Barbier, F. F., Dun, E. A., Kerr, S. C., Chabikwa, T. G. & Beveridge, C. A. An update on the signals controlling shoot branching. Trends Plant Sci. 24, 220–236 (2019).
3. Line 276-312: the authors analyzed the promotors in the SWEET genes while the data indicated the cold temperature did not affect most of the genes; however, in figure 7, it looks like the temperature influenced the genes pattern, which might need a link or explanation. Moreover, the author claimed a similar expression pattern in the same period of different regions (fig 7C), but the data presented here did not fit. Please check it carefully.
4. In the qRTPCR figure, there is no statistical analysis that makes it difficult to assess the claim. Also, probably use two colors to distinguish the two genotypes.
5. Line 337-349/Line 359-366, this section was introduced at the beginning of the text (or at least similar), which looked wordy. Please revise it or shorten it.
6. I felt the discussion part is too dense to read which might make the reader confused. I would suggest the authors shorten and focus on “main” findings.

Experimental design

no comment

Validity of the findings

no comment

---

## Round 0.2 · Major Revisions

Dear authors,

Thank you kindly for all of the improvements made to this revised version of your manuscript as they have improved on the standard of your original submission greatly.

However, as you will see from the attached and annotated PDF of your revised manuscript, post my review of your revised manuscript, considerable additional revisions are required to further improve the standard of your manuscript. Without these additional revisions being made, I cannot consider your manuscript any further.

Specific issues still outstanding are:

(1) Extensive English language issues persist throughout the manuscript. Please use the attached annotated PDF as a template to identify all revisions being required / requested.

(2) The Discussion section remains very poor. Yes the degree of claims made in the Discussion of your original manuscript version have been lessened considerably, however, the text that you have left in the revised Discussion is now largely simply a repeat of the reporting of your obtained results, most of which were already reported on in the Results section. The Discussion is to be used to interpret your obtained results, and this is done by using existing literature to either confirm or contrast the results of your study with those previously reported by others. Please completely rewrite the Discussion taking this required approach.

(3) The headings of numerous Figures require improvement to properly convey to the reader the data being reported on in each Figure.

(4) An appropriate level of detail is missing from numerous Figure legends. More detail is therefore required in order for a reader to be able to fully appreciate the data being reported on in each manuscript Figure without having to rely on the text of the manuscript to find the associated information.

(5) Please carefully consider the formatting of each Figure to ensure that the major finding/s being reported on in each Figure carriers maximum impact.

Considering the degree of work I am requesting as part of this round of revision please do take your time with this process to ensure that all raised concerns are adequately addressed.

Kind regards,
Andrew.

---

## Round 0.3 · Minor Revisions

Dear authors,

A large number of English language issues remain in the revised version of your manuscript.

Please see the annotated version of your revised study to identify each of these English language issues which remain outstanding.
Please carefully revise the current version of your manuscript and then resubmit it for further consideration.

Yes this version of the manuscript has improved considerably, however, additional work remains before I can consider this manuscript further.

Kind regards,
Andrew.

---

## Round 0.4 · Minor Revisions

Dear authors,

There still exists a number of typographical errors within the most current revised version of your manuscript.

Please use the attached annotated PDF of my review of your manuscript and correct each outstanding issue.

Please return the finalised version of your manuscript to me for my final decision.

Regards,
Andrew.

---

## Round 0.5 · accepted · Accept

Dear authors,

Thank you for continuing to cooperate with me throughout the publication process.

The latest version of your manuscript is much improved on the originally submitted version.

Considering these improvements, your manuscript is now at a standard suitable for publication acceptance in PeerJ. I congratulate you on this achievement.

Please do be aware that a very small number of grammatical issues do remain in the text of this version of the manuscript, so please do take the opportunity to correct these issues in the next stages of the publication process.

Well done to the authorship team.
Kind regards,
Andrew.